# TAM: Temporal Adaptive Module for Video Recognition

## Abstract

Video data is with complex temporal dynamics due to various factors such as camera motion, speed variation, and different activities. To effectively capture this diverse motion pattern, this paper presents a new temporal adaptive module (**TAM**) to generate video-specific temporal kernels based on its own feature maps. TAM proposes a unique two-level adaptive modeling scheme by decoupling dynamic kernel into a location sensitive importance map and a location invariant aggregation weight. The importance map is learned in a local temporal window to capture short term information, while the aggregation weight is generated from a global view with a focus on long-term structure. TAM is a principled module and could be integrated into 2D CNNs to yield a powerful video architecture (TANet) with a very small extra computational cost. The extensive experiments on Kinetics-400 and Something-Something datasets demonstrate that the TAM outperforms other temporal modeling methods consistently, and achieves the state-of-the-art performance under the similar complexity.

## 1 Introduction

Deep learning has brought great progress for various recognition tasks in image domain, such as image classification (Krizhevsky et al., 2012; He et al., 2016), object detection (Ren et al., 2017), and instance segmentation (He et al., 2017). The key to these successes is to devise flexible and efficient architectures that are capable of learning powerful visual representations from large-scale image datasets (Deng et al., 2009). However, deep learning research progress in video understanding is relatively more slowly, partially due to the high complexity of video data. The core technical problem in video understanding is to design an effective temporal module, that is expected to be able to capture complex temporal structure with high flexibility, while yet to be of low computational consumption for processing high dimensional video data efficiently.

3D Convolutional Neural Networks (3D CNNs) (Ji et al., 2010; Tran et al., 2015) have turned out to be mainstream architectures for video modeling (Carreira & Zisserman, 2017; Feichtenhofer et al., 2019; Tran et al., 2018; Qiu et al., 2017). The 3D convolution is a natural extension over its 2D counterparts and provides a learnable operator for video recognition. However, this simple extension lacks specific consideration about the temporal properties in video data and might as well lead to high computational cost. Therefore, recent methods aim to improve 3D CNNs from two different aspects by combining a lightweight temporal module with 2D CNNs to improve efficiency (e.g., TSN (Wang et al., 2016), TSM (Lin et al., 2019)), or designing a dedicated temporal module to better capture temporal relation (e.g., Nonlocal Net (Wang et al., 2018b), ARTNet (Wang et al., 2018a), STM (Jiang et al., 2019)). However, how to devise a temporal module with high efficiency and strong flexibility still remains to be an unsolved problem in video recognition. Consequently, we aim at advancing the current video architectures along this direction.

In this paper, we focus on devising a principled adaptive module to capture temporal information in a more flexible way. In general, we observe that video data is with extremely complex dynamics along the temporal dimension due to factors such as camera motion and various speed. Thus 3D convolutions (temporal convolutions) might lack enough representation power to describe motion diversity by simply employing a fixed number of **video invariant** kernels. To deal with such complex temporal variations in videos, we argue that **adaptive temporal kernels** for each video are effective and as well necessary to describe motion patterns. To this end, as shown in Figure 1,

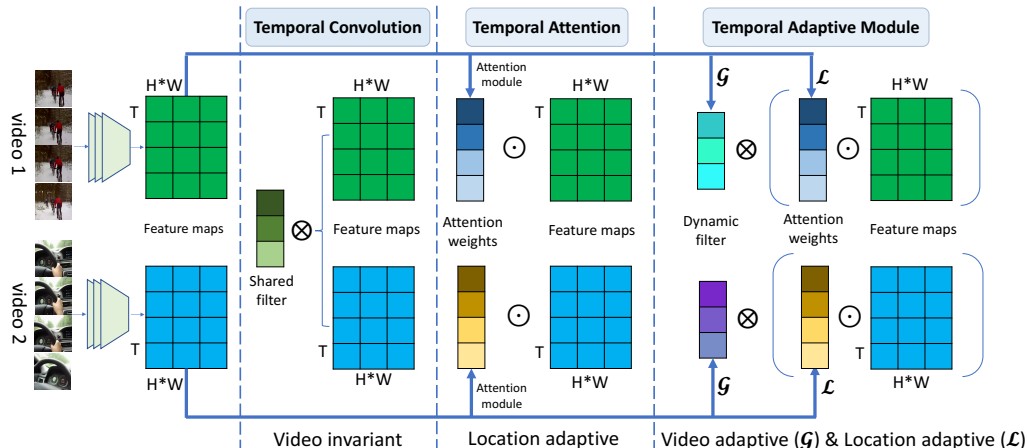

Figure 1: **Temporal module comparison:** The standard *temporal convolution* shares weights among videos and may lack the flexibility to handle video variations due to the diversity of videos. The *temporal attention* learns position sensitive weights by assigning varied importance for different time without any temporal interaction, and may ignore the long-range temporal dependencies. Our proposed *temporal adaptive module* (TAM) presents a two-level adaptive scheme by learning the local importance weights for location adaptive enhancement and the global kernel weights for video adaptive aggregation. ⊙ denotes attention operation, and ⊗ denotes convolution operation.

we present a two-level adaptive modeling scheme to decompose this video specific temporal kernel into a location sensitive *importance map* and a location invariant (also video adaptive) *aggregation kernel*. This unique design allows the location sensitive importance map to focus on enhancing discriminative temporal information from a local view, and enables the location invariant aggregation weights to capture temporal dependencies guided by a global view of the input video sequence.

Specifically, the design of *temporal adaptive module* (TAM) strictly follows two principles: high efficiency and strong flexibility. To ensure our TAM with a low computational cost, we first squeeze the feature map by employing a global spatial pooling, and then establish our TAM in a channel-wise manner to keep the efficiency. Our TAM is composed of two branches: a local branch ($\mathcal{L}$) and a global branch ($\mathcal{G}$). As shown in Fig. 2, TAM is implemented in an efficient way. The local branch employs temporal convolutions to produce the location sensitive importance maps to discriminate the local feature, while the global branch uses fully connected layers to produce the location invariant kernel for temporal aggregation. The importance map generated by a local temporal window focuses on short-term motion modeling and the aggregation kernel using a global view pays more attention to the long-term temporal information. Furthermore, our TAM could be flexibly plugged into the existing 2D CNNs to yield an efficient video recognition architecture, termed as TANet.

We validate the proposed TANet on the task of action classification in video recognition. Particularly, we first study the performance of the TANet on the Kinetics-400 dataset. We demonstrate that our TAM is better at capturing temporal information than other several counterparts, such as temporal pooling, temporal convolution, TSM (Lin et al., 2019), and Non-local block (Wang et al., 2018b). Our TANet is able to yield a very competitive accuracy with the FLOPs similar to 2D CNNs. We further test our TANet on the motion dominated dataset of Something-Something, where the state-of-the-art performance is also achieved.

## 2 RELATED WORKS

Video understanding is a core topic in the field of computer vision. At early stage, a lot of traditional methods (Le et al., 2011; Kläser et al., 2008; Sadanand & Corso, 2012; Willems et al., 2008) have designed various hand-crafted features to encode the video data, but these methods are too inflexible when generalized to other video tasks. Recently, since the rapid development of video understanding has been much benefited from deep learning methods (Krizhevsky et al., 2012; Simonyan & Zisserman, 2015; He et al., 2016), especially in video recognition, a series of CNNs-based methods

were proposed to learn spatiotemporal representation, and the differences with our method will be clarified later. Furthermore, our work also relates to dynamic convolution and attention in CNNs.

**CNNs-based Methods for Action Recognition.** Since the deep learning method has been wildly used in the image tasks, there are many attempts (Karpathy et al., 2014; Simonyan & Zisserman, 2014; Wang et al., 2016; Zhou et al., 2018; He et al., 2019; Lin et al., 2019) based on 2D CNNs devoted to modeling the video clips. In particular, Wang et al. (2016) used the frames sparsely sampled from the whole video to learn the long-range information by aggregating scores after the last fully-connected layer. Lin et al. (2019) shifted the channels along the temporal dimension in an efficient way, which yields a good performance with 2D CNNs. By a simple extension from spatial domain to spatiotemporal domain, 3D convolution (Ji et al., 2010; Tran et al., 2015) was proposed to capture the motion information encoded in video clips. Due to the release of large-scale Kinetics dataset (Kay et al., 2017), 3D CNNs (Carreira & Zisserman, 2017) were wildly used in action recognition. Its variants (Qiu et al., 2017; Tran et al., 2018; Xie et al., 2018) decomposed the 3D convolution into a spatial 2D convolution and a temporal 1D convolution to learn the spatiotemporal features. And Feichtenhofer et al. (2019) designed a network with dual paths to learn the spatiotemporal features and achieved a promising accuracy in video understanding.

The methods aforementioned all share a common insight that they are video invariant and ignore the inherent temporal diversities in videos. As opposed to these methods, we design a two-level adaptive modeling scheme by decomposing the video specific operation into a location sensitive excitation and a location invariant convolution with adaptive kernel for each video clip.

**Attention in Action Recognition.** The local branch in TAM mostly relates to SENet (Hu et al., 2018). But the SENet learned modulation weights for each channel of feature maps. Several methods (Liu et al., 2019b; Diba et al., 2018) also resorted to the attention to learn more discriminative features in videos. Different from these methods, the local branch keeps the temporal information to learn the location sensitive importances. Wang et al. (2018b) designed a non-local block which can be seen as self-attention to capture long-range dependencies. Our TANet captures the long-range dependencies by simply stacking more TAM, and keep the efficiency of networks.

**Dynamic Convolutions.** Jia et al. (2016) first proposed the dynamic filters on the tasks of video and stereo prediction, and designed a convolutional encoder-decoder as filter-generating network. Several works (Yang et al., 2019; Chen et al., 2020) in image tasks attempted to generate aggregation weights for a set of convolutional kernels, and then produce a dynamic kernel. Our motivation are different from these methods. We aim to use this temporal adaptive module to deal with temporal variations in videos. Specifically, we design an efficient form to implement this temporal dynamic kernel based on input feature maps, which is critical for understanding the video content.

## 3 METHOD

### 3.1 THE OVERVIEW OF TEMPORAL ADAPTIVE MODULE

As we discussed in Sec.1, video data typically exhibit the complex temporal dynamics caused by many factors such as camera motion and speed variations. Therefore, we aim to tackle this issue by introducing a temporal adaptive module (TAM) with video specific kernels, unlike the sharing convolutional kernel in 3D CNNs. our TAM could be easily integrated into the existing 2D CNNs (e.g., ResNet) to yield a video network architecture, as shown in Figure 2. We will give an overview of TAM and then describe its technical details.

Formally, let $X \in \mathbb{R}^{C \times T \times H \times W}$ denote the feature maps for a video clip, where $C$ represents the number of channels, and $T, H, W$ are its spatiotemporal dimensions. For efficiency, our TAM only focus on temporal modeling and the spatial pattern is expected to captured by 2D convolutions. Therefore, we first employ a global spatial average pooling to squeeze the feature map as follows:

$$\hat{X}_{c,t} = \phi(X)_{c,t} = \frac{1}{H \times W} \sum_{i,j} X_{c,t,j,i}, \tag{1}$$

where $c, t, j, i$ is the index of different dimensions (in channel, time, height and width), and $\hat{X} \in \mathbb{R}^{C \times T}$ aggregates the spatial information of $X$. For simplicity, we here use $\phi$ to denote the function

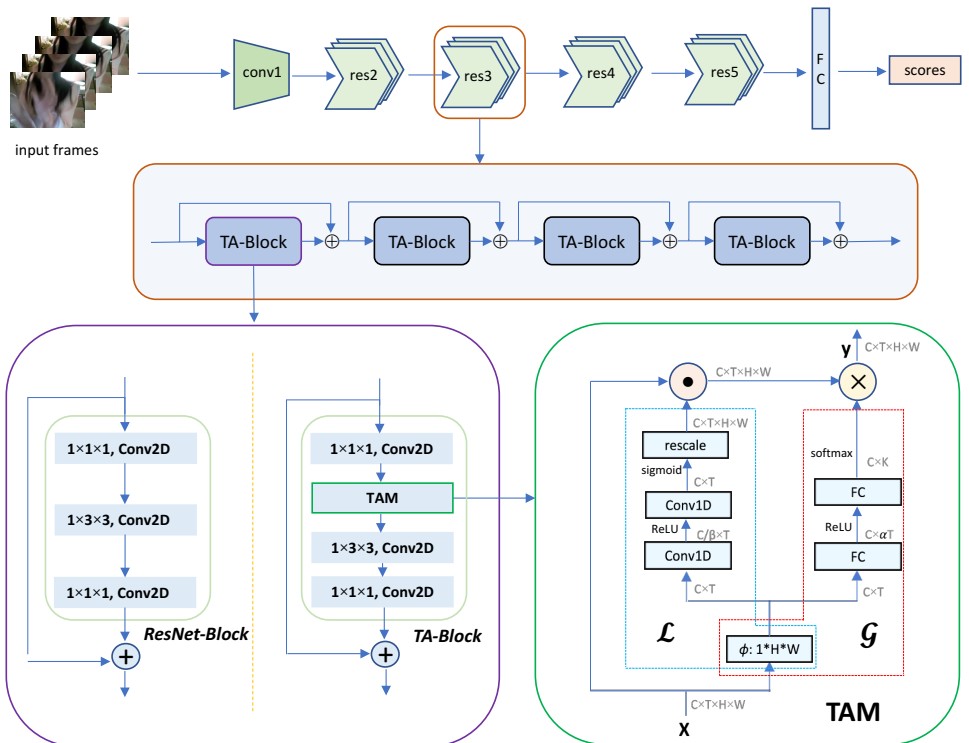

Figure 2: **The overall architecture of TANet:** ResNet-Block *vs*. TA-Block. The whole workflow of temporal adaptive module (TAM) in the lower right shows how it works. The shape of tensor has noted after each step. ⊕ denotes element-wise addition, ⊙ denotes element-wise multiplication, and ⊗ denotes convolution operator. The symbols appeared in figure will be explained in Sec. 3.1 .

that aggregates the spatial information. Our proposed temporal adaptive module is established based on this squeezed 1D temporal signal for a high efficiency.

Our TAM is composed of two branches: a local branch $\mathcal{L}$ and a global branch $\mathcal{G}$, which aims to learn a location sensitive importance map to enhance discriminative features and then produces the location invariant weights to adaptively aggregate temporal information in a convolutional manner. More specifically, the TAM is formulated as follows:

$$Y = \mathcal{G}(X) \otimes (\mathcal{L}(X) \odot X), \tag{2}$$

where ⊗ denotes convolutional operator and ⊙ denotes element-wise multiplication. It is worth noting that these two branches focus on different aspects of temporal information, where the local branch tries to capture the short term information to attend important features by using a temporal convolution, while the global branch aims to incorporate long-range temporal structure to guide adaptive temporal aggregation with fully connected layers. Disentangling kernel learning procedures into local and global branches turns out to be an effective way in experiments. The two branches will be introduced in the following sections.

## 3.2 LOCAL BRANCH IN TAM

As discussed above, the local branch is location sensitive and aims to leverage short-term temporal dynamics to perform video specific operation. Given that the short-term information varies slowly along the temporal dimension, it is thus required to learn a location sensitive importance map to discriminate the local temporal semantics.

As shown in Figure 2, we build the local branch with a sequence of temporal convolutional layers with ReLU non-linearity As the goal of local branch is to capture short term information, we set the kernel size $K$ as 3 to learn importance map solely based on a local temporal window. To control the model complexity, the first $\mathrm{Conv1D}$ followed by BN (Ioffe & Szegedy, 2015) reduces the number of

channels from $C$ to $\frac{C}{\beta}$. Then, the second $\mathrm{Conv1D}$ with a sigmoid activation yields the importance weights $V \in \mathbb{R}^{C \times T}$ which are sensitive to temporal location. Finally, the temporal excitation is formulated as follows:

$$Z = \mathrm{F}_{\mathrm{rescale}}(V) \odot X = \mathcal{L}(X) \odot X, \tag{3}$$

where $\odot$ denotes the element-wise multiplication and $Z \in \mathbb{R}^{C \times T \times H \times W}$. To match size of $X$, $\mathrm{F}_{\mathrm{rescale}}(V)$ rescales the $V$ to $\hat{V} \in \mathbb{R}^{C \times T \times H \times W}$ by replicating in spatial dimension.

### 3.3 GLOBAL BRANCH IN TAM

The global branch, also termed as location invariant branch, focuses on generating an adaptive kernel based on long-term temporal information. It incorporates global context information and learns to produce the location invariant and also video adaptive convolution kernel for dynamic aggregation.

**Learning the Adaptive Kernels.** We here opt to generate the dynamic kernel for each video clip and aggregate temporal information in a convolutional manner. To simply this procedure and as well as preserve high efficiency, The adaptive convolution will be applied in a channel-wise manner. In this sense, we expect our learned adaptive kernel only considers the temporal relations without taking channel correlation into account. Thus, our TAM would not change the number of channels of input feature maps, and the learned adaptive kernel convolves the input feature maps in a channel-wise manner. More formally, for the $c^{th}$ channel, the adaptive kernel is learned as follows:

$$\Theta_c = \mathcal{G}(X)_c = \mathrm{softmax}(\mathcal{F}(\mathbf{W_2}, \delta(\mathcal{F}(\mathbf{W_1}, \phi(X)_c)))), \tag{4}$$

where $\Theta_c \in \mathcal{R}^K$ is generated adaptive kernel (aggregation weights) for $c^{th}$ channel, $K$ is the adaptive kernel size, $\delta$ denotes the activation function *ReLU*. The adaptive kernel is also learned based on the squeezed feature map $\hat{X}_c \in \mathbb{R}^T$ without taking the spatial structure into account for modeling efficiency. But different with the local branch, we use fully connected ($fc$) layers $\mathcal{F}$ to learn the adaptive kernel by leveraging long-term information. The learned adaptive kernel with the global receptive field, thus could aggregate temporal features guided by the global context. To increase the modeling capabilities of the global branch, we stack two $fc$ layers and the learned kernel is normalized with a softmax function to yield a positive aggregation weight. The learned aggregation weights $\Theta = \{\Theta_1, \Theta_2, ..., \Theta_C\}$ will be employed to perform video adaptive convolution.

**Temporal Adaptive Aggregation.** Before introducing the adaptive aggregation, we can look back on how a vanilla temporal convolution aggregates the spatio-temporal visual information:

$$Y = W \otimes X, \tag{5}$$

Where $W$ is the weights of convolution kernel and has no concern with input video samples in inference. We argue this fashion ignores the temporal dynamics in videos, and thus propose a video adaptive aggregation to model video clips:

$$Y = \mathcal{G}(X) \otimes X, \tag{6}$$

where $\mathcal{G}$ can be seen as a kernel generator function. Kernel generated by $\mathcal{G}$ can perform adaptive convolution, but is still location invariant and shared cross temporal dimension. To address this issue, the local branch produces $Z$ with location sensitive importance map. The whole procedures can be expressed as follows:

$$Y_{c,t,j,i} = \mathcal{G}(X) \otimes Z = \Theta \otimes Z = \sum_k \Theta_{c,k} \cdot Z_{c,t+k,j,i}, \tag{7}$$

where $\cdot$ denotes the scalar multiplication, $Y \in \mathbb{R}^{C \times T \times H \times W}$ is the output feature maps.

In summary, TAM presents an adaptive module with a unique two-step aggregation scheme, where the location sensitive excitation and location invariant aggregation all derive from input features, but focus on capturing different structures (i.e., short-term and long-term temporal structure).

### 3.4 EXEMPLAR: TANET

We here intend to describe how to instantiate the TANet. Temporal adaptive module as a novel temporal modeling method can endow the existing 2D CNNs with a strong ability to model different

temporal structures in video clips. In practice, TAM only causes limited computing overhead, but obviously improves the performance on different types of datasets.

ResNets (He et al., 2016) are employed as backbones to verify the effectiveness of TAM. As illustrated in Fig. 2, the TAM is embedded into ResNet-Block after the first Conv2D, which easily turns the vanilla ResNet-Block into TA-Block. This fashion will not excessively alter the topology of networks and can reuse the weights of ResNet-Block. Supposing we sample T frames as an input clip, the scores of T frames after $fc$ will be aggregated by average pooling to yield the clip-level scores. No temporal downsampling is performed before $fc$ layer. The extensive experiments are conducted in Sec. 4 to demonstrate the flexibility and efficacy of TANet.

### 3.4.1 DISCUSSIONS

We have noticed that the structure of local branch is similar to the SENet (Hu et al., 2018) and STC (Diba et al., 2018). The first obvious difference is our local branch does not squeeze the temporal information. We thus use temporal 1D convolution as a basic layer, instead of using $fc$ layer. Two-layer design only seeks to make a trade-off between non-linear fitting capability and model complexity. Furthermore, the local branch provides the location sensitive information, and thus addresses the issue that the global branch is insensitive to temporal location.

TSN (Wang et al., 2016), TSM (Lin et al., 2019), etc. only aggregate the temporal features with a fixed scheme, but our TAM can yield the video specific weights to adaptively aggregate the temporal features in different stages. In the extreme cases, our global branch in TAM can degenerate into TSN when dynamic kernel weights $\Theta$ is learned to equal to $[0, 1, 0]$. From another perspective, if the kernel weights $\Theta$ is set to $[1, 0, 0]$ or $[0, 0, 1]$, global branch can be turned into TSM. It seems that our TAM theoretically provides a more general and flexible form to model the video data. When it refers to 3D convolution (Ji et al., 2010), all input samples share the same convolution kernel without being aware of the temporal diversities in videos as well. In addition, our global branch essentially performs a video adaptive convolution whose filter has size $1 \times k \times 1 \times 1$, while each filter in a normal 3D convolution has size $C \times k \times k \times k$, where $C$ is the number of channels and k denotes the receptive field. Thus our method is more efficient than 3D CNNs. Unlike some current dynamic convolution (Chen et al., 2020; Yang et al., 2019), TAM is more flexible, and can directly generate the kernel weights to perform video adaptive convolution.

## 4 EXPERIMENTS

In this section, we elaborately study the effectiveness of TANet on several standard benchmarks. The training recipe and inference protocol is described in **Appendix (A.1)**.

### 4.1 DATASETS

Our experiments are conducted on three large scale datasets, namely, Kinetics-400 (Kay et al., 2017) and Something-Something (Sth-Sth) V1&V2 (Goyal et al., 2017). Kinetics-400 contains ∼300k video clips with 400 human action categories. The videos in Kinetics-400, trimmed from raw YouTube videos, are around 10s. We here train models on training set (∼240k video clips), and test models on validation set (∼20k video clips). The Sth-Sth datasets focus on fine-grained action, which contains a series of pre-defined basic actions interacted with daily objects. The Sth-Sth V1 comprises ∼86k video clips in training set and ∼12k video clips in validation set. Sth-Sth V2 is an updated version of Sth-Sth V1, which contains ∼169k video clips in training set and ∼25k video clips in validation set. They both have 174 action categories.

### 4.2 EXPLORATION STUDIES ON KINETICS-400

The exploration studies are performed on Kinetics-400 to investigate different aspects of TANet. The ResNet architecture we used is the same with He et al. (2016). Our TANet replaces all ResNet-Blocks with TA-Blocks by default.

Table 1: Ablation studies on Kinetics-400. All models use ResNet50 as backbone.

(a) Studying on parameter choices of $\alpha$ and $\beta$.

| setting | Frames | Top-1 | Top-5 |
|---|---|---|---|
| $\alpha=1$, $\beta=4$ | 8 | 75.63% | 92.10% |
| $\alpha=2$, $\beta=4$ | 8 | **76.09%** | **92.30%** |
| $\alpha=4$, $\beta=4$ | 8 | 75.72% | 92.14% |
| $\alpha=2$, $\beta=2$ | 8 | 75.91% | 92.38% |
| $\alpha=2$, $\beta=4$ | 8 | **76.09%** | **92.30%** |
| $\alpha=2$, $\beta=8$ | 8 | 75.63% | 92.20% |

(b) Trying larger temporal receptive fields of $\Theta$.

| Kernel | Frames | Top-1 | Top-5 |
|---|---|---|---|
| K=3 | 8 | 76.09% | 92.30% |
| K=5 | 8 | 75.62% | 92.14% |
| K=3 | 16 | 76.87% | 92.88% |
| K=5 | 16 | 77.19% | 93.17% |

Table 2: Study on the effectiveness of TAM. All models use ResNet50 as backbone and take 8 frames with sampling stride 8 as inputs. To be consistent with testing, the FLOPs are calculated with spatial size $256 \times 256$. All methods share the same training setting and inference protocol.

| Models | FLOPs (of single view) | Params | Top-1 | Top-5 |
|---|---|---|---|---|
| C2D | 42.95G | 24.33M | 70.2% | 88.9% |
| C2D-Pool | 42.95G | 24.33M | 73.1% | 90.6% |
| C2D-TConv | 53.02G | 28.10M | 73.3% | 90.7% |
| TSM (Lin et al., 2019) | 42.95G | 24.33M | 74.1% | 91.2% |
| TEINet (Liu et al., 2019b) | 43.01G | 25.11M | 74.9% | 91.8% |
| I3D$_{3\times1\times1}$ (Wang et al., 2018b) | 62.55G | 32.99M | 74.3% | 91.6% |
| NL C2D (Wang et al., 2018b) | 64.49G | 31.69M | 74.4% | 91.5% |
| Global branch | 43.00G | 24.33M | 74.9% | 91.7% |
| Local branch | 43.00G | 25.59M | 73.3% | 90.7% |
| Global branch + SE (Hu et al., 2018) | 43.02G | 24.65M | 75.4% | 92.0% |
| TANet-R | 43.02G | 25.59M | 76.0% | 92.2% |
| TANet | 43.02G | 25.59M | **76.1%** | **92.3%** |

**Parameter choices.** We use different combinations of $\alpha$ and $\beta$ to figure out the optimal hyper-parameters in TAM. The TANet is instantiated as in Fig. 2. TANet with $\alpha = 2$ and $\beta = 4$ achieves the highest performance shown in Table 1a, which will be applied in following experiments.

**Temporal receptive fields.** We also try to increase the temporal receptive fields for learned kernel $\Theta$ in the global branch. From the Table 1b, it seems the larger $K$ is beneficial to the accuracy when TANet takes more sampled frames as inputs. On the other hand, it even degenerates the performance of TANet when sampling 8 frames. In our following experiments, the $K$ will be set to 3.

The results in Table 1 have revealed that our TANet is insensitive to these Hyper-parameters, which can save a lot of time to find optimal settings in practice. We also provide other more exploration studies in **Appendix (A.2)**, and show how these hyper-parameters impact the performance of TANet.

### 4.3 COMPARISON WITH OTHER TEMPORAL MODULES

As a principled temporal operator, we intend to describe its competitive counterparts and then make fair comparisons with TAM in Table 2. The optimal configurations studied in Sec. 4.2 will be employed in the following experiments. The default inference protocol samples 10 clips $\times$ 3 crops to evaluate the performance of each model.

**2D ConvNet (C2D).** We use ResNet50 as backbone to build 2D ConvNet. The 2D ConvNet focuses on learning the spatial clues, which operates on each frame independently without any temporal interaction before the last $fc$ layer.

**C2D-Pool.** To probe into the impacts of temporal fusion, C2D-Pool utilizes the average pooling layer whose kernel size is $K \times 1 \times 1$ to perform temporal fusion without any temporal downsampling, which can be built by easily replacing all TAMs in network with pooling layers.

**C2D-TConv.** We also replace each TAM with a standard temporal convolution, and this comparison is able to demonstrate the importance of adaptive modeling in temporal aggregation.

Table 3: Extending to other backbones. All models share the same inference protocol, e.g., 10 clips $\times$ 3 crops.

| Models | ShuffleNet V2 | | MobileNet V2 | | Inception V3 | | ResNet-50 | | ResNext-50 | |
|---|---|---|---|---|---|---|---|---|---|---|
| | Top-1 | Top-5 | Top-1 | Top-5 | Top-1 | Top-5 | Top-1 | Top-5 | Top-1 | Top-5 |
| w/o TAM | 62.1% | 84.3% | 64.1% | 85.6% | 71.4% | 89.8% | 70.7% | 88.9% | 70.1% | 88.8% |
| with TAM | 67.3% | 87.6% | 71.6% | 90.1% | 75.6% | 92.0% | 76.1% | **92.3%** | **76.4%** | 92.0% |
| $\Delta Acc.$ | *+ 5.2%* | *+ 3.3%* | *+ 7.5%* | *+ 4.5%* | *+ 4.2%* | *+ 2.2%* | *+ 5.4%* | *+ 3.4%* | *+ 6.3%* | *+ 3.2%* |

There are some competitive methods based on C2D, i.e., **TSM** (Lin et al., 2019) and **TEINet** (Liu et al., 2019b). The new methods focus on designing a lightweight temporal module and can be inserted into the 2D CNN for efficiently capturing temporal information.

**Inflated 3D ConvNet (I3D).** I3D (Carreira & Zisserman, 2017) is most frequently used models in action recognition. In our implementation, we inflate the first $1 \times 1$ kernel in ResNet-Block to $3 \times 1 \times 1$, which can provide more fair comparisons with our TANet. Following the (Wang et al., 2018b), we use I3D$_{3 \times 1 \times 1}$ to denote this variant.

The aforementioned methods share the same temporal modeling scheme with a fixed pooling or convolution operation. As shown in Table 2, our method yield a superior performance that higher than C2D by 5.9% accuracy, and even outperforms I3D$_{3 \times 1 \times 1}$ (76.1% vs. 74.3%), which exhibits the fixed schemes for modeling videos may be insufficient to learn the temporal clues. As the extra convolutions in C2D-TConv might destroy the ImageNet pretrained weights, C2D-TConv even achieves a degenerated performance compared with I3D$_{3 \times 1 \times 1}$. In addition, compared with other temporal counterparts in video recognition, TANet only brings a small portion of FLOPs and parameters.

**Non-local C2D (NL C2D).** The non-local block was proposed to capture the long-range dependencies in videos. The preferable settings with 5 non-local blocks mentioned in (Wang et al., 2018b) are employed to compare with TANet. As seen in Table 2, TANet achieves higher accuracy than NL C2D (76.1% vs. 74.4%). In addition, TANet is more efficient than NL C2D. TANet only has 43G FLOPs of single view and 25.6M parameters.

To study the each part of temporal adaptive module, we separately validate the **Global branch** and **Local branch**. Furthermore, **Global branch + SE** uses global branch with SE module (Hu et al., 2018) to compare with TANet, which can prove the complementarity of local branch and global branch. TANet has achieved the highest accuracy among these models, which proves the efficacy of each part of TAM and as well as the strong complementarity between local branch and global branch. As opposed to Equ. 2, TANet-R combines the global and local branch in reverse order:

$$Y = \mathcal{L}(X) \odot (\mathcal{G}(X) \otimes X), \tag{8}$$

We found TANet is slightly better than TANet-R.

**Generalization to other Backbones.** One critical issue that we need to figure out is the generalization of our proposed method extending to other backbone networks. To this end, we extend the TAM from ResNet to other well known classification backbones, like ShuffleNet V2 (Ma et al., 2018), MobileNet V2 (Sandler et al., 2018), Inception V3 (Szegedy et al., 2016) and ResNeXt-50 (Xie et al., 2017). From the Table 3, we can observe that the backbone networks equipped with our TAM outperform their C2D baselines by a large margin, which strongly exhibits the powerful generalization as well as the huge potential of our temporal adaptive module.

### 4.4 COMPARISONS WITH THE STATE OF THE ART

**Comparisons on Kinetics-400.** Table 4 shows the state-of-the-art results on Kinetics-400. Our method, as an adaptive modeling scheme, has achieved competitive performance compared with other models. TANet-50 with 8-frame also outperforms SlowFast (Feichtenhofer et al., 2019) by 0.5% when using similar FLOPs per view. The 16-frame TANet only uses 4 clips and 3 crops for evaluation such that it provides higher inference efficiency and more fair comparisons with other models. It is worth noting that our 16-frame TANet-50 is still more accurate than 32-frame NL I3D by 2.2%. As ip-CSN (Tran et al., 2019) is pretrained on Sports-1M (Karpathy et al., 2014), it also achieves the promising accuracy with deeper backbone, i.e., ResNet152. Furthermore, TAM is com-

Table 4: Comparisons with the state-of-the-art methods on Kinetics-400. As described in Feichtenhofer et al. (2019), the GFLOPs (of a single view) × the number of views (temporal clips with spatial crops) represents the model complexity. The GFLOPs are calculated with spatial size $256 \times 256$. The models in gray rows represent it is unjust to directly compare with our method.

| Methods | Backbones | Training Input | GFLOPs×views | Top-1 | Top-5 |
|---|---|---|---|---|---|
| TSN (Wang et al., 2016) | InceptionV3 | 3×224×224 | 3×250 | 72.5% | 90.2% |
| ARTNet (Wang et al., 2018a) | ResNet18 | 16 ×112×112 | 23.5×250 | 70.7% | 89.3% |
| S3D-G (Xie et al., 2018) | InceptionV1 | 64×224×224 | 71×30 | 74.7% | 93.4% |
| I3D (Carreira & Zisserman, 2017) | InceptionV1 | 64×224×224 | 108×N/A | 72.1% | 90.3% |
| R(2+1)D (Tran et al., 2018) | ResNet34 | 32×112×112 | 152×10 | 74.3% | 91.4% |
| NL I3D (Wang et al., 2018b) | ResNet50 | 32×224×224 | N/A | 74.9% | 91.6% |
| NL I3D (Wang et al., 2018b) | ResNet50 | 128×224×224 | 282×30 | 76.5% | 92.6% |
| ip-CSN (Tran et al., 2019) | ResNet50 | 8×224×224 | 1.2×10 | 70.8% | -% |
| TSM (Lin et al., 2019) | ResNet50 | 16×224×224 | 65×30 | 74.7% | 91.4% |
| TEINet (Liu et al., 2019b) | ResNet50 | 16×224×224 | 86×30 | 76.2% | 92.5% |
| bLVNet-TAM-24×2 | bLResNet50 | 48×224×224 | 93×9 | 73.5% | 91.2% |
| SlowOnly (Feichtenhofer et al., 2019) | ResNet50 | 8×224×224 | 42×30 | 74.8% | 91.6% |
| SlowFast (Feichtenhofer et al., 2019) | ResNet50 | (4+32)×224×224 | 36×30 | 75.6% | 92.1% |
| TANet-50 | ResNet50 | 8×224×224 | 43×30 | 76.1% | 92.3% |
| TANet-50 | ResNet50 | 16×224×224 | 86×12 | **76.9%** | **92.9%** |
| X3D-XL (Feichtenhofer, 2020) | - | 16×312×312 | 48×30 | 79.1% | 93.9% |
| ip-CSN (Tran et al., 2019) | ResNet152 | 32 ×224×224 | 83×30 | 79.2% | 93.8% |
| SlowFast+NL (Feichtenhofer et al., 2019) | ResNet101 | (16+64)×224×224 | 234×30 | 79.8% | 93.9% |

patible with existing video frameworks like SlowFast. Specifically, our TAM is more lightweight than $3 \times 1 \times 1$ convolution when taking the same number of frames as inputs, but yields a better performance. TAM thus can easily replace the $3 \times 1 \times 1$ convolution in SlowFast to achieve higher accuracy with lower computational costs. It seems that X3D has achieved a great success in video recognition. In other ways, X3D was searched by massive computing resource and can not be easily obtained in any situation. Although our method does not beat all state-of-the-art methods equipped with deeper networks, TAM as a lightweight operator can enjoy the benefits from more powerful backbones and video frameworks. In general, the proposed TANet makes a good practice on adaptively modeling the temporal relations in videos.

**More Results and Analysis.** The results on Sth-Sth V1 & V2 is presented in **Appendix (A.3)** in which TAM also achieves a competitive performance compared with other methods. To have more intuitive understandings of temporal adaptive module, we also visualize the learned kernels in **Appendix (A.4)**, which are expected to provide more insights for TAM. As shown in Figure 4, the diversities in our learned kernels have shown that the complex dynamics are indeed existing in videos, and learning temporal clues in an adaptive scheme has proven to be effective yet reasonable.

## 5 CONCLUSION

In this paper, we have presented a novel temporal modeling operator, i.e., temporal adaptive module (TAM), to capture complex motion information in videos and built a powerful video architecture (TANet). Our TAM is able to yield a video-specific kernels with the combination of a local importance map and a global aggregation weight. The local and global branches designed in TAM are helpful to capture temporal structure by different views and contribute to making temporal modeling more effective and robust. As demonstrated on the Kinetics-400, the networks equipped with TAM are better than the existing temporal modules in action recognition, which confirms the efficacy of our TAM in video temporal modeling. TANet also achieved the state-of-the-art performance on the motion dominated datasets of Sth-Sth V1&V2.

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

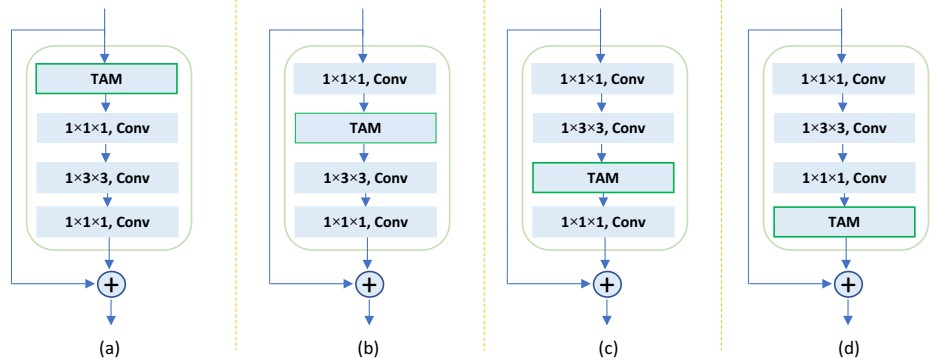

Figure 3: **The four styles of TA-Block.** The (b) is actually the model we used in the main text.
.

Brandon Yang, Gabriel Bender, Quoc V. Le, and Jiquan Ngiam. Condconv: Conditionally parameterized convolutions for efficient inference. In Hanna M. Wallach, Hugo Larochelle, Alina Beygelzimer, Florence d'Alché-Buc, Emily B. Fox, and Roman Garnett (eds.), *Advances in Neural Information Processing Systems 32: Annual Conference on Neural Information Processing Systems 2019, NeurIPS 2019, 8-14 December 2019, Vancouver, BC, Canada*, pp. 1305–1316, 2019.

Bolei Zhou, Alex Andonian, Aude Oliva, and Antonio Torralba. Temporal relational reasoning in videos. In *Computer Vision - ECCV*, pp. 831–846, 2018.

Mohammadreza Zolfaghari, Kamaljeet Singh, and Thomas Brox. ECO: efficient convolutional network for online video understanding. In *Computer Vision - ECCV*, pp. 713–730, 2018.

## A  APPENDIX

### A.1  IMPLEMENTATION DETAILS

**Training.** In our experiments, we only train the models using 8 frames and 16 frames as inputs. On Kinetics-400, following the practice in Wang et al. (2018b), The frames are sampled from 64 consecutive frames in the video. On Sth-Sth V1&V2, we employ the uniform sampling strategy in TSN (Wang et al., 2016) to train TANet. We first resize the shorter side of frames to 256, and apply the multi-scale cropping and randomly horizontal flipping as data augmentation. The cropped frames are resized to $224 \times 224$ for training the networks. The batch size is set to 64. Our models are initialized by ImageNet pre-trained weights to reduce the training time. Specifically, on Kinetics-400, the epoch for training is 100. The initial learning rate is set 0.01 and divided by 10 at 50, 75, 90 epoch. We use SGD with a momentum of 0.9 and a weight decay of 1e-4 to train TANet. On Sth-Sth V1&V2, We train models with 50 epochs. The learning rate starts at 0.01 and divided by 10 at 30, 40, 45 epoch. We use a momentum of 0.9 and a weight decay of 1e-3 to address the issue of overfitting.

**Testing.** We apply different inference schemes to fairly compare with other state-of-the-art models. On kinetics-400, we resize the shorter to 256 and take 3 crops of $256 \times 256$ to cover the spatial dimensions. In the temporal dimension, we uniformly sample 10 clips for 8-frame models and 4 clips for 16-frame models. The final video-level prediction is yielded by averaging the scores of all spatio-temporal views. On Sth-Sth V1, we scale the shorter side of frames to 256 and use center crop of $224 \times 224$ for evaluation. On Sth-Sth V2, we employ similar evaluation protocols to Kinetics, but only uniformly sample 2 clips.

### A.2  MORE EXPLORATION STUDIES ON KINETICS-400

**TAM in the different position.** Table 5a tries to study the effects of TAM in different position. TANet-a, TANet-b, TANet-c, and TANet-d denotes the TAM is inserted before the first convolution, after the first convolution, after the second convolution, and after the last convolution in the block,

Table 5: Ablation studies on Kinetics-400. All models use ResNet50 as backbone.

(a) Where to insert TAM into TA-Block.

| model | Frames | Top-1 | Top-5 |
|-------|--------|-------|-------|
| TANet-a | 8 | 75.95% | 92.18% |
| TANet-b | 8 | **76.09%** | **92.30%** |
| TANet-c | 8 | 75.75% | 92.13% |
| TANet-d | 8 | 75.20% | 91.78% |

(b) The number of TA-Blocks inserted into ResNet50.

| stages | Frames | Blocks | Top-1 | Top-5 |
|--------|--------|--------|-------|-------|
| $res_5$ | 8 | 3 | 74.12% | 91.45% |
| $res_{4-5}$ | 8 | 9 | 75.15% | 92.04% |
| $res_{3-5}$ | 8 | 13 | 75.90% | 92.22% |
| $res_{2-5}$ | 8 | 16 | **76.09%** | **92.30%** |

respectively. These four styles are graphically presented in Fig. 3. The style-(b) in Fig. 2 actually is TANet-b, which has a slighter advantage than other styles as shown in Table 5a. The TANet-b will be abbreviated as TANet by default.

**The number of TA-Blocks.** To make a trade-off between performance and efficiency, we gradually add more TA-Blocks into ResNet. As shown in Table 5b, we find that the performance is nearly saturated when adding more than 9 TA-Blocks into network. The $res_{2-5}$ achieves the highest performance and will be used in our experiments.

A.3    COMPARISONS ON STH-STH V1 & V2

Table 6: Comparisons with the state-of-the-art methods on Sth-Sth V1. The models only taking RGB frames as inputs are listed in table. To be consistent with testing, we use spatial size 224×224 to compute the FLOPs.

| Methods | Backbones | Pre-train | Frames | FLOPs | Top-1 | Top-5 |
|---------|-----------|-----------|--------|-------|-------|-------|
| TSN-RGB (Wang et al., 2016) | BNInception | ImgNet | $8f$ | 16G | 19.5% | - |
| TRN-Multiscale (Zhou et al., 2018) | BNInception | ImgNet | $8f$ | 33G | 34.4% | - |
| S3D-G (Xie et al., 2018) | Inception | ImgNet | $64f$ | 71.38G | 48.2% | 78.7% |
| ECO (Zolfaghari et al., 2018) | BNIncep+Res18 | K400 | $16f$ | 64G | 41.6% | - |
| $ECO_{En}$Lite (Zolfaghari et al., 2018) | BNIncep+Res18 | K400 | $92f$ | 267G | 46.4% | - |
| TSN (Wang et al., 2016) | ResNet50 | ImgNet | $8f$ | 33G | 19.7% | 46.6% |
| I3D (Wang & Gupta, 2018) | ResNet50 | ImgNet+K400 | $32f \times 2$ | 306G | 41.6% | 72.2% |
| NL I3D (Wang & Gupta, 2018) | ResNet50 | ImgNet+K400 | $32f \times 2$ | 334G | 44.4% | 76.0% |
| NL I3D+GCN (Wang & Gupta, 2018) | ResNet50+GCN | ImgNet+K400 | $32f \times 2$ | 606G | 46.1% | 76.8% |
| TSM (Lin et al., 2019) | ResNet50 | ImgNet | $8f$ | 33G | 45.6% | 74.2% |
| TSM (Lin et al., 2019) | ResNet50 | ImgNet | $16f$ | 65G | 47.2% | 77.1% |
| $TSM_{En}$ (Lin et al., 2019) | ResNet50 | ImgNet | $16f + 8f$ | 98G | 49.7% | 78.5% |
| TAM (Fan et al., 2019) | ResNet50 | ImgNet | $8f$ | - | 46.1% | -% |
| bLVNet-TAM (Fan et al., 2019) | ResNet50 | Sth-Sth V2 | $32f$ | 48G | 48.4% | 78.8% |
| GST (Luo & Yuille, 2019) | ResNet50 | ImgNet | $8f$ | 30G | 47.0% | 76.1% |
| GST (Luo & Yuille, 2019) | ResNet50 | ImgNet | $16f$ | 59G | 48.6% | 77.9% |
| TEINet Liu et al. (2019b) | ResNet50 | ImgNet | $8f$ | 33G | 47.4% | -% |
| TEINet Liu et al. (2019b) | ResNet50 | ImgNet | $16f$ | 66G | 49.9% | -% |
| TANet | ResNet50 | ImgNet | $8f$ | 33G | 46.5% | 75.8% |
| TANet | ResNet50 | ImgNet | $16f$ | 66G | 47.6% | 77.7% |
| $TANet_{En}$ | ResNet50 | ImgNet | $8f + 16f$ | 99G | **50.6%** | **79.3%** |

**Comparisons on Sth-Sth V1 & V2**. As shown in Table 6, our method achieves state-of-the-art accuracy comparing with other models on Sth-Sth V1. For fair comparisons, the Table 6 only reports the results taking a single clip with a center crop as inputs. $TANet_{En}$ is higher than $TSM_{En}$ equipped with same backbone (Top-1: 50.6% vs. Top-1: 49.7%). We also conduct the experiments on Sth-Sth V2. V2 has more video clips than V1, which can further unleash the full capabilities of TANet without suffering the overfitting. Following the common practice in Lin et al. (2019), TANets use 2 clips with 3 crops to evaluate the accuracy. As shown in Table 7, our models have achieved the state-of-art performance on Sth-Sth V2. As a result, the $TANet_{En}$ yields a competitive accuracy (Top-1: 66.0%) compared with current SOTA results. The results on Sth-Sth V1 & V2

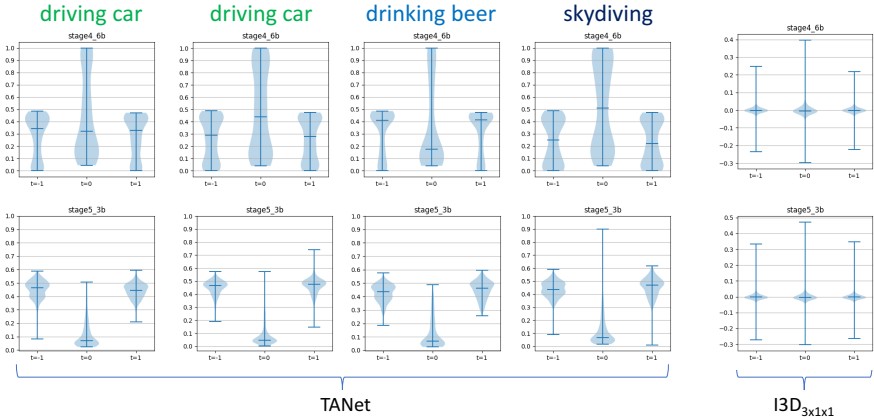

Figure 4: The statistics of kernel weights training on Kinetics-400, which plots the distributions in different temporal offsets ($t \in \{-1, 0, 1\}$). Each filled area in violinplot represents the entire data range, where has noted the minimum, the median and the maximum. The first four columns in the left figure are the distributions of learned kernels in TANet. In the fifth column, we also visualize the filters of $3 \times 1 \times 1$ kernel in I3D$_{3 \times 1 \times 1}$ to compare with the TANet. The **stage4_6b** denotes the kernel comes from the 6th block in stage4.

have demonstrated that our method is also good at modeling the fine-grained and temporal-related video clips.

Table 7: Comparisons with the SOTA on Sth-Sth V2.

| Methods | Backbones | Pre-train | frames×clips×crops | Top-1 | Top-5 |
|---|---|---|---|---|---|
| TRN (Zhou et al., 2018) | BNInception | ImgNet | $8f \times 2 \times 3$ | 48.8% | 77.6% |
| TSM (Lin et al., 2019) | ResNet50 | ImgNet | $8f \times 2 \times 3$ | 59.1% | 85.6% |
| TSM (Lin et al., 2019) | ResNet50 | ImgNet | $16 \times 2 \times 3$ | 63.4% | 88.5% |
| TSM$_{RGB+Flow}$ (Lin et al., 2019) | ResNet50 | ImgNet | $(16 + 16) \times 2 \times 3$ | 66.0% | 90.5% |
| CPNet (Liu et al., 2019a) | ResNet50 | ImgNet | $24f \times 16 \times 16$ | 57.7% | 84.0% |
| GST (Luo & Yuille, 2019) | ResNet50 | ImgNet | $8f \times 1 \times 1$ | 61.6% | 87.2% |
| GST (Luo & Yuille, 2019) | ResNet50 | ImgNet | $16f \times 1 \times 1$ | 62.6% | 87.9% |
| bLVNet-TAM (Fan et al., 2019) | ResNet50 | Sth-Sth V2 | $32f \times 1 \times 1$ | 61.7% | 88.1% |
| TEINet Liu et al. (2019b) | ResNet50 | ImgNet | $8f \times 1 \times 1$ | 61.3% | -% |
| TEINet Liu et al. (2019b) | ResNet50 | ImgNet | $16f \times 1 \times 1$ | 62.1% | -% |
| TANet | ResNet50 | ImgNet | $8f \times 1 \times 1$ | 60.5% | 86.2% |
| TANet | ResNet50 | ImgNet | $8f \times 2 \times 3$ | 62.7% | 88.0% |
| TANet | ResNet50 | ImgNet | $16 \times 1 \times 1$ | 62.5% | 87.6% |
| TANet | ResNet50 | ImgNet | $16 \times 2 \times 3$ | 64.6% | 89.5% |
| TANet$_{En}$ | ResNet50 | ImgNet | $(8f+16f) \times 2 \times 3$ | **66.0%** | **90.1%** |

## A.4 VISUALIZATIONS OF LEARNED KERNEL

To understand the behavior of TANet, we visualize the distribution of kernel $\Theta$ generated by global branch in the last block of stage4 and stage5. For clear contrast, the kernel weights in I3D$_{3 \times 1 \times 1}$ at the same stages are also visualized to find more insights. As depicted in Fig. 4, we find that the learned kernel $\Theta$ has an evident character: the shapes and scales of distribution are more diverse than I3D$_{3 \times 1 \times 1}$. Since all video clips share the same kernels in I3D$_{3 \times 1 \times 1}$, it causes the kernel weights clusters together excessively. As opposed to temporal convolution, even modeling the same action in different videos, TAM can generate the kernel with slightly different distributions. Taking driving car as an example, the shapes of the distribution shown in Fig. 4 are similar to each other but the medians of distributions are not equal. For different actions like drinking beer and skydiving, the shapes and medians of distributions are greatly varied. Even the same action in different videos, TAM would learn a different distribution of kernel weights. Concerning that the motion patterns in different videos may share varied inherence, it is necessary to employ an adaptive scheme when modeling video sequences.

To probe into the effects on learning kernels in the different stages, the visualized kernels are further chosen in stage4_6b and stage5_3b, respectively. The videos are randomly selected from Kinetics-400 and Sth-Sth V2 to show the diversities in different video datasets. As depicted in Fig. 5 and Fig. 6, We can observe that the distributions of importance map $V$ in the local branch are smoother than the kernel $\Theta$ in the global branch, and the local branch pays different attention to each video

when modeling the temporal relations. Futhermore, our learned kernels visualized in figures have exhibited the clear differences between two datasets (Kinetics-400 vs. Sth-Sth V2). This fact is in line with our prior knowledge that there is an obvious domain gap between two datasets. The Kinetics-400 mainly focuses on appearance and Sth-Sth V2 is a motion dominated dataset. However, this point can not be easily inferred from the kernels in I3D$_{3\times1\times1}$, because the overall distributions of kernels in I3D$_{3\times1\times1}$ on two datasets show minor differences.

We visualize the histogram of kernel weights in the global branch to intuitively show the insightful patterns between kernel weights and visual content. As observing the Fig. 7 and Fig. 8, we found the distribution of kernel is associated with the motion magnitude. The smaller magnitude of motion in video leads to a relatively lower weight at $t = 0$, and the larger magnitude of motion or the actions severely related to scenes may cause the higher weight at $t = 0$. It is worth noting that the weight at t=0 usually plays a dominate role in learning the spatio-temporal representations. These findings are expected to provide more insights for designing the temporal module in video recognition.

Generally, the diversities in our learned kernels have demonstrated that the diversities are indeed existing in videos, and it is reasonable to learn spatiotemporal representation in an adaptive scheme. These findings are again in line with our motivation claimed in the paper.

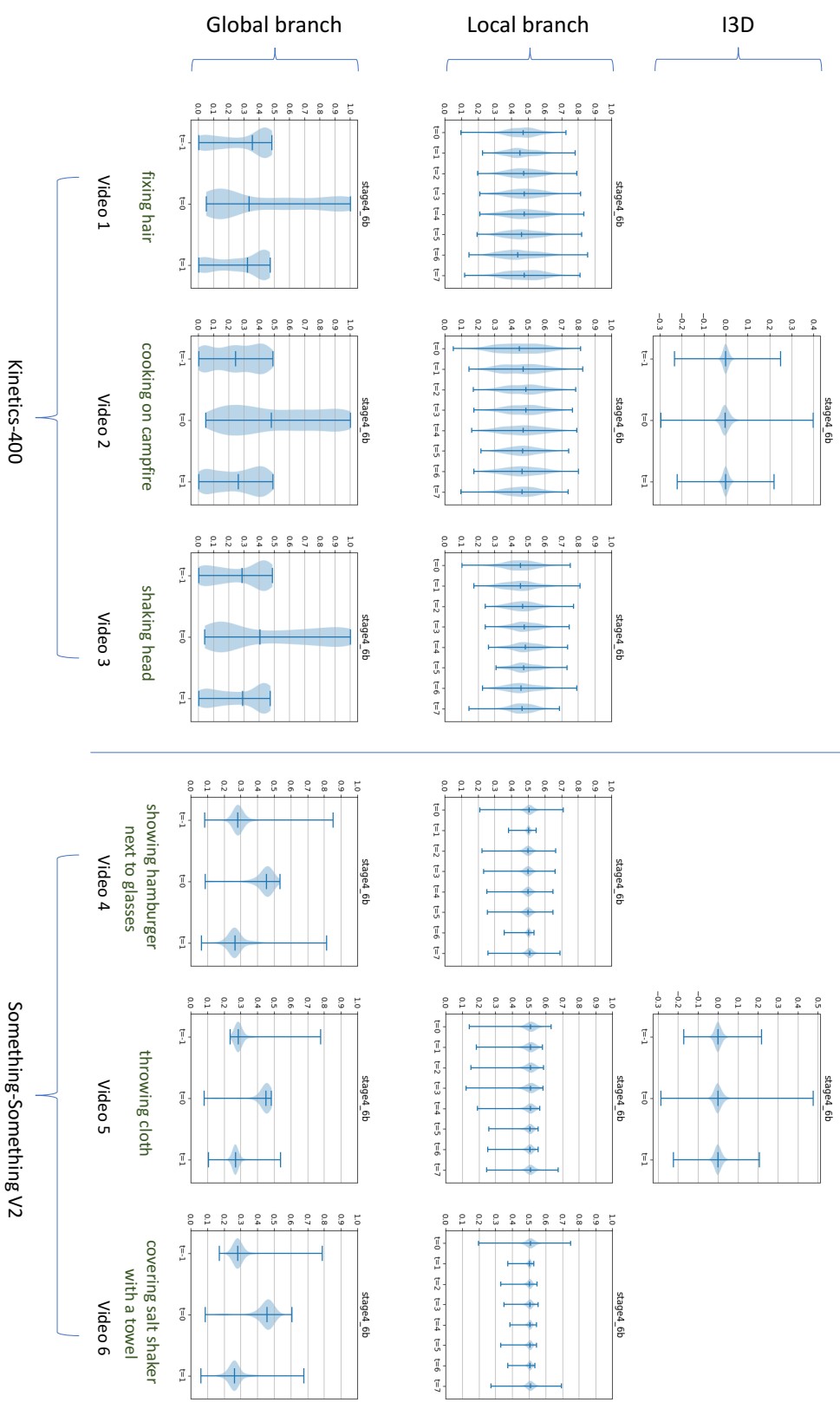

Figure 5: The distribution of learned kernel $V$ and $\Theta$ in the stage4_6b.

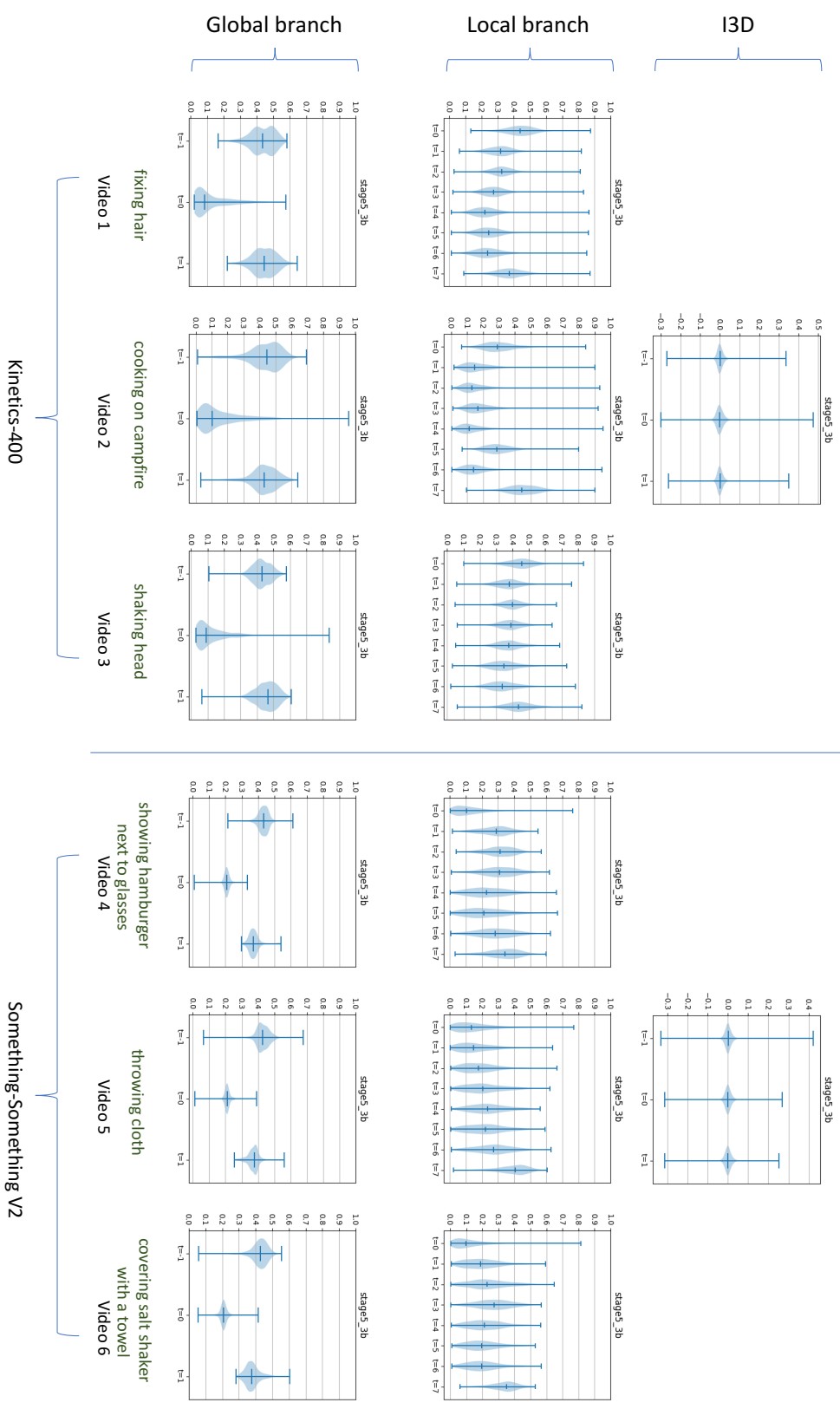

Figure 6: The distribution of learned kernel $V$ and $\Theta$ in the stage5_3b.

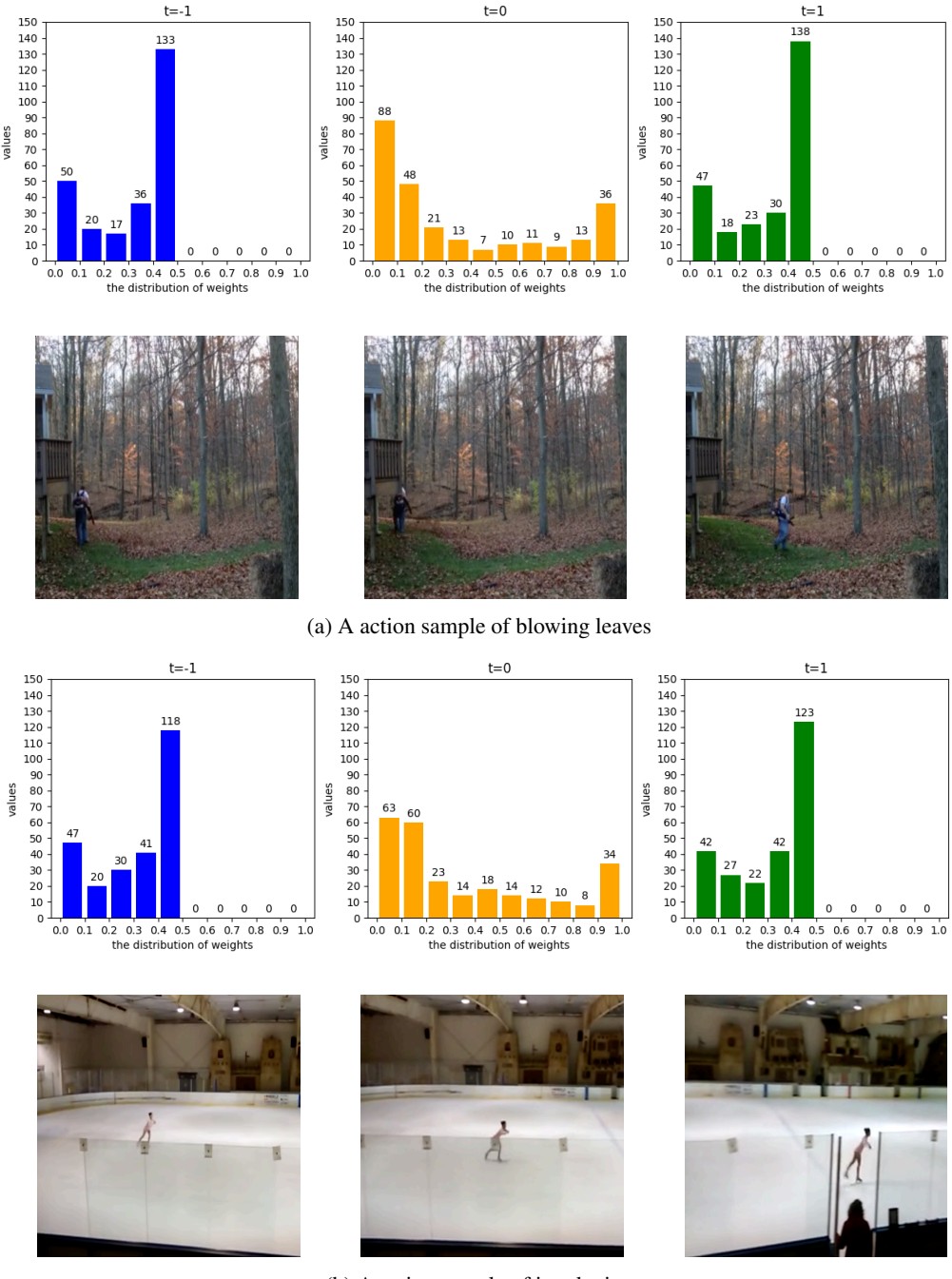

Figure 7: The visualization for the videos which contain small magnitude of motion. The moving persons appeared in videos are usually far away from camera. Video clips are sampled from validation set of Kinetics-400, and kernel Θ is selected from stage4_6b.

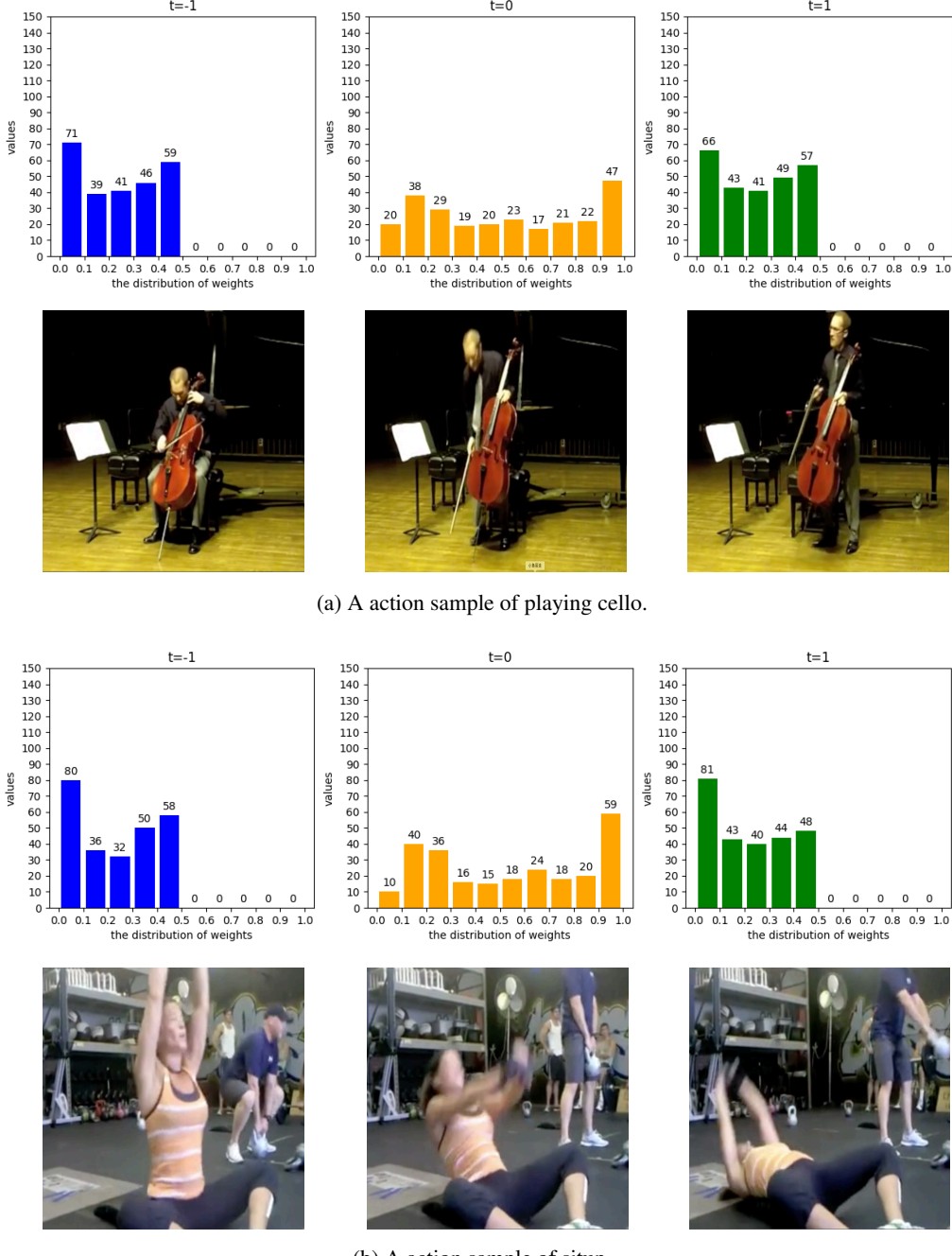

(a) A action sample of playing cello.

(b) A action sample of situp.

Figure 8: The visualization for the videos with large magnitude of motion. The moving persons in videos are usually closed to camera. Video clips are sampled from validation set of Kinetics-400, and kernel $\Theta$ is selected from stage4_6b.

