# OpenReview forum: "TAM: Temporal Adaptive Module for Video Recognition"
_ICLR.cc/2021/Conference — Reject_

### Official Review · AnonReviewer4 · 2020-10-28
**The interpretability of the distributions of the important map V and the kernel \Theta.**

**Rating:** 6
**Confidence:** 4

**Review:**

This paper presents a new temporal adaptive module (TAM) to generate video-specific temporal kernels based on its own feature maps. TAM proposes a unique two-level adaptive modeling scheme by decoupling dynamic kernel into a location sensitive importance map and a location invariant aggregation weight. The importance map is learned in a local temporal window to capture short term information, while the aggregation weight is generated from a global view with a focus on long-term structure.

The global branch aims to incorporate long-range temporal structure to guide adaptive temporal aggregation with fully connected layers. The adaptive kernel (aggregation weights) is learned based on long-term temporal information, and incorporates global context information and learns to produce the location invariant and also video adaptive convolution kernel for dynamic aggregation. The visualization of the statistics of kernel weights shows that the shapes and scales of distribution are more diverse and data-adaptive. It is indeed reasonable to learn spatiotemporal representation in an adaptive scheme.

However, the paper can be improved further.
1.	Some SOTA performances are ignored selectively, such as some performances of Slowfast[1].
[1] Christoph Feichtenhofer, Haoqi Fan, Jitendra Malik, and Kaiming He. Slowfast networks for video recognition. In ICCV, pp. 6201–6210, 2019
Maybe the proposed method does not achieve the best performance, but it is necessary to compare with the SOTA methods completely. Deep analysis can help the readers get the core contribution of the paper.
2.	The TAM only focus on the temporal modeling. This may limit the final performance of the network. If the idea can be extended to the spatial-temporal field, it will be more valuable.
3.	The authors give the distributions of the important map V in the local branch and the kernel \Theta in the global branch. Since these V and \Theta are all data-dependent, it is not unexpected that the distributions have larger diversities than the traditional I3D’s temporal kernel. If the authors can try to explore the relationship between the distributions and the characteristic of some action samples, beyond the diversity visualization of distributions, it will be more convincing.
4.	Fig. 1 is difficult to understand. It is not straightforward to figure out how the attention weights or kernel weights are learned. Arrows in Fig. 1 are also confusing. Some indicate names, while some indicate feature flows.

---

> ### Author Response · Authors · 2020-11-23
> **Response to reviewer 4**
>
> Thanks for your comments. Your advice is helpful for us to improve our submission, and potentially provide some new insight to our method as well. The related codes with pretrained models will be released as soon as possible
>
>
>
> **Q1.** Some SOTA performances are missed.
>
> R1. Thanks for your suggestion. We have put some missing results (e.g., SlowFast [1], ip-CSN [2]) into the Table 4. Although our method does not beat these state-of-the-art methods using deeper backbone networks, TAM as a lightweight operator provide a new perspective to learn the temporal clues in video, which can enjoy the benefits from more powerful backbones and video frameworks. To this end , we have examined the TAM with other well-known classification backbones as well, which can be found in Table 3 of the resubmitted paper. More detailed discussions have put into Section 4.3 and 4.4 of the revised paper.
>
>
>
> **Q2.** If the idea can be extended to the spatio-temporal field, it will be more valuable.
>
> **R2.** During rebuttal period, we extend our TAM from the temporal field to the spatial-temporal field as your suggestion. It will increase a large amount of computation overhead if we directly generate a full k\*d\*d spatio-temporal kernel. Consequently, a new scheme is proposed by decoupling the k\*d\*d kernel into k\*1\*1, 1\*d\*1 and 1\*1\*d kernels. Technically, the temporal k\*1\*1 kernel is generated in the same way as before. The 1\*d\*1 and 1\*1\*d employ a similar design as global branch in original paper, but it reduces the feature dimension by a factor of 2 after first fc layer for efficiency. The training settings and inference protocols are kept consistent with its original counterpart. In our experiments, k=3 and d=3. Results are as follows:
>
> |  Model   | Top-1 | Top-5 |
> | :------: | :---: | :---: |
> |  TANet   | 75.5% | 92.1% |
> | TANet-ST | 75.2% | 92.0% |
>
> TANet-ST represents the method that extends the TAM from temporal field to spatio-temporal field. Since the 1\*d\*1 and 1\*1\*d are generated based on spatial feature map, in the testing, the spatial resolution (H\*W) of input clips should be aligned with training phase, namely, H\*W=224*224. The spatio-temporal version of TAM leads to a little drop in accuracy (~0.3%).
>
> We analyze that the 2D backbone network plays a dominant role in learning the spatial semantics , and the extra 1\*d\*1 and 1\*1\*d dynamic convolutions contribute little to the final performance. It perhaps make the model fall into the issuse of overfitting.
>
>
>
> **Q3.** Exploring the relationship between the distributions and the characteristic of some action samples.
>
> **R3.** We visualize the histogram of kernel weights in the global branch to intuitively show the insightful patterns between kernel weights and visual content. As observing the Figure 7 and Figure 8, we found the distribution of kernel is associated with the motion magnitude. The smaller magnitude of motion in video leads to a relatively lower weight at $t=0$, and the larger magnitude of motion or the actions severely related to scenes may cause the higher weight at $t=0$. It is worth noting that the weight at t=0 usually plays a dominate role in learning the spatio-temporal representations. We hope these findings can provide more insights for designing the temporal module in video recognition. These analyses have been  put into Appendix (A.4).
>
>
>
> **Q4.** It is not straightforward to figure out how the attention weights or kernel weights are learned, and arrows in Fig. 1 are also confusing.
>
> **R4.** For your concerns, we have revised the Figure 1 to make it more accessible. In the revised figure, the arrows in Figure 1 only denote where the attention weights and kernel weights are derived.
>
>
>
>
>
> [1]. Christoph Feichtenhofer, Haoqi Fan, Jitendra Malik, and Kaiming He. Slowfast networks for video recognition. ICCV 2019.
>
> [2]. Du Tran, Heng Wang, Matt Feiszli, and Lorenzo Torresani. Video classification with channel-separated convolutional networks. ICCV 2019.

---

### Official Review · AnonReviewer1 · 2020-10-29
**The experiments are not thorough enough.**

**Rating:** 4
**Confidence:** 5

**Review:**

This paper proposes a temporal adaptive module for video recognition. Specifically, it decouples dynamic kernel into a location sensitive importance map and a location invariant aggregation weight, which can be plugged into existing 2D CNNs to yield a powerful video architecture with small extra computational cost. The experiments conducted on several datasets demonstrate the effectiveness of the proposed method.

Paper Strength
(1)	The proposed method develops a simple but effective module for video recognition, achieving good performance on various kinds of datasets.
(2)	The proposed two-level adaptive modeling scheme is effective to describe motion patterns. Specifically, it decomposes the video specific temporal kernel into a location sensitive importance map and a location invariant aggregation kernel.

Paper Weakness
(1)	Figure 1 is confusing. Why the dimension of the green filter is 3*1*1?
(2)	In Figure 2, TAM is used before the 1*1*3 convolutional layer in the ResNet block. What about the influence of TAM at different locations? Is there any insight for this design?
(3)	Moreover, what if we exchange the order of the local branch and the global branch?
(4)	In Section 3.4, the authors claim that the proposed temporal adaptive module can be plugged into existing 2D CNNs with a strong ability to model different temporal structures in video clips. However, only the ResNet-50 backbone is verified in the experiment. More experiments with other backbones such as VGG and Inception should be added.
(5) In Table 3, the proposed method is not compared to the state-of-the-art X3D method [*] on the Kinetics-400 dataset.

[*] X3D: Expanding Architectures for Efficient Video Recognition.

Summary
This paper proposes a temporal adaptive module for action recognition. The proposed module is straight-forward and obtains good performance on different datasets. The experiments are not thorough enough to demonstrate that the proposed module can be plugged into different 2D CNNs with good performance.

---

> ### Author Response · Authors · 2020-11-23
> **Response to reviewer 1**
>
> Thanks for your valuable comments. Your concerns are fully addressed in the following and the main result will be incorporated into the final version. Our code and pre-trained models will be also released very soon.
>
> **Q1.** Figure 1 is confusing. Why the dimension of the green filter is 3\*1\*1?
>
> **R2.** The kernel size of a temporal convolution should be k\*d\*d (k is temporal size and d is spatial size), and d is equal to 1 in our paper. In our paper, the spatial dimension (H\*W) in Figure 1 is compressed for better visualization. Meanwhile, I have revised Figure 1 to make it more accessible.
>
>
>
> **Q2.** In Figure 2, TAM is used before the 1\*3\*3 convolutional layer in the ResNet block. What about the influence of TAM at different locations?
>
> **R2.** In Appendix A.2, we have studied the effect of TAM at different positions in the ResNet Bottleneck. We surprisedly found that different variations have brought only a slight fluctuation in performance. The results in Table 5 have demonstrated that our TAM is not so sensitive to these settings.  As the number of channels of input feature maps will be reduced after first convolution in bottleneck, we use the setting in Figure 2 by default to keep a high inference efficiency, .
>
>
>
> **Q3.** What if we exchange the order of the local branch and the global branch?
>
> **R3.** The experimental results are as follows:
>
> | Models  | Top-1 | Top-5 |
> | :-----: | :---: | :---: |
> |  TANet  | 76.1% | 92.3% |
> | TANet-R | 76.0% | 92.2% |
>
> The model, termed as TANet-R, exchanges the order of the local branch and the global branch in Equ 2. We found TANet is only slightly better than TANet-R.  This result has also been put into Table 2 in the revised version.
>
>
>
> **Q4.** More experiments with other backbones should be added.
>
> **R4.** According to your suggestion, we have conducted the experiments that incorporate the TAM into other  backbones such as ShuffleNet [1], MobileNet [2], Inception [3] and ResNeXt [4]. As shown in the following table:
>
> |   Models   | ShuffleNet V2 | MobileNet V2 | Inception V3 | ResNet-50 | ResNext-50 |
> | :--------: | :-----------: | :----------: | :----------: | :-------: | ---------- |
> |  w/o TAM   |     62.2%     |    64.1%     |    71.4%     |   70.7%   | 70.1%      |
> |  with TAM  |     67.3%     |    71.6%     |    75.6%     |   76.1%   | 76.4%      |
> | Detal Acc. |    + 5.1%     |    + 7.5%    |    + 4.2%    |  + 5.4%   | + 6.3%     |
>
>
>
> We can observe that the backbone networks equipped with our TAM outperform their C2D baselines by a large margin, which exhibits the good generalization of our temporal adaptive module.
>
> These results have been put into Table 3 in our revised paper, and the detailed discussions can be found in the Sec. 4.3 of paper as well.
>
>
>
> **Q5.** In Table 3, the proposed method is not compared to the X3D [5]  on the Kinetics-400 dataset.
>
> **R5.** Thanks for your reminder, the missing method has been put into Table 4.  X3D uses network architecture search strategy to discover the optimal setting for 3D CNNs, while our TAM focuses on designing new temporal modules. Our TAM is on an orthogonal direction and complementary to X3D , in sense that our TAM could be incorporated into the X3D architecture and X3D could perform NAS based on TAM.
>
>
>
> [1]. Ningning Ma, Xiangyu Zhang, Hai-Tao Zheng, and Jian Sun.  Shufflenet V2:  practical guidelinesfor efficient CNN architecture design. ECCV 2018.
>
> [2]. Mark Sandler, Andrew G. Howard, Menglong Zhu, Andrey Zhmoginov, and Liang-Chieh Chen. Mobilenetv2: Inverted residuals and linear bottlenecks. CVPR 2018.
>
> [3]. Christian Szegedy, Vincent Vanhoucke, Sergey Ioffe, Jonathon Shlens, and Zbigniew Wojna. Rethinking the inception architecture for computer vision. CVPR 2016.
>
> [4]. Saining Xie, Ross B. Girshick, Piotr Dollar, Zhuowen Tu, and Kaiming He. Aggregated residual transformations for deep neural networks. CVPR 2017.
>
> [5]. Christoph Feichtenhofer. X3D: expanding architectures for efficient video recognition. CVPR 2020.

---

### Official Review · AnonReviewer2 · 2020-10-29
**Review of the manuscript TAM: Temporal Adaptive Module for Video Recognition**

**Rating:** 8
**Confidence:** 4

**Review:**

*********
Summary Of The Manuscript:
*********
The manuscript addresses the problem of Video Recognition one of the applications of Computer Vision. Due to various complex temporal dynamics of video data (Camera Motion, Speed, etc.), to capture the vast information, the author presents a novel Temporal Adaptive Module (TAM) for generating kernels based on the temporal feature maps. In addition, these feature maps are a combination of local and global features and as an exemplar, the author presents an architecture - TANet by incorporating their temporal operator. Together with a variety of experiments on standard benchmark for Video Recognition: Kinetics - 400 and Something-Something, the author showcases that for the task of Video Recognition compared to existing temporal operators, TAM's performance is fairly consistent and better and archives State-of-the-art with similar complexity in their exemplar architecture - TANet.

*********
Strength Of The Manuscript:
*********
++ Novelty

- The task formulation is concise, convincing, and novel. A seemingly reasonable approach has been proposed in this manuscript for the task of Video Recognition. Compared to the existing baseline and recent approaches, the proposed architecture - TANet achieves SOTA results.
- To the best of my knowledge, the incorporation of two branches - Local and Global branch makes the whole operator efficient and flexible for adaptation in the frameworks by stacking them to capture more complex information. Thus the concept of the TAM is convincing to capture complex temporal information.
- In addition, the exemplar showcased by the authors - TANet has been created by incorporating TAM in the existing 2-Dimensional CNNs to capture vast information which proves that the proposed module/operator is flexible enough and can be adapted to different frameworks/architectures for better performance.

++ Clarity

- The manuscript is written in an excellent way to provide a brief insight into TAM. Especially Subsection 3.2 and 3.2 provide a good in-depth description of how the local and global branch works effectively in a joint manner to capture the short and long-term complex temporal information.
- The manuscript also clearly describes the improvements and adequately contextualizes the contributions in such a way that it makes a good starting point for a novice reader.

++ Evaluation

- The experiments are sufficient and convincing. This new operator and exemplar TANet shows improved performance in nearly all cases on the datasets
- The experimental evaluations demonstrate the effectiveness of the proposed architecture and showcase its practical value.
- Also, an ablation analysis demonstrates to gain an understanding of where the performance benefits have been obtained such as receptive fields and parameter choices.

*********
Weakness Of The Manuscript:
*********
Overall, currently at this stage, this is a very good and strong manuscript in my entire batch. I like the simplicity and wide applicability of the proposed operator, especially the incorporation of local and global branches and adaptive aggregation. Thus I do not have any major weakness issues after reading the manuscript several times. Detailed literature review, a complete overview of each component, and detailed experiments and ablation studies helps to give a good insight into the manuscript. I found this paper pretty solid and have not able to found concerns relating to the proposed work.

*********
Justification Of The Review:
*********
Overall, happy with the current version of the manuscript. As mentioned earlier, I like the simplicity and wide applicability of the proposed module, and the architecture and setup details are provided in such a manner that it is very easy to convert into code in some timeframe. Detailed literature review, a complete overview of each component, and detailed experiments and ablation studies help to understand the author's work. Finally, I think the paper is pretty solid and thus I prefer to give a rating of 8 currently.

---

> ### Author Response · Authors · 2020-11-23
> **Responses to reviewer 2**
>
> Thanks for your positive comments. More experiments such as using other well-known classification backbones and related analyses have been added into the revised paper to show the potential and efficacy of TAM. To  facilitate the development of  video recognition,  the related code and pre-trained models will be released very soon.

---

### Decision · Program_Chairs · 2021-01-07
**Final Decision**

**Decision:**

Reject

**Comment:**

This paper proposes a temporal module for video representation learning, which is a combination of temporal attention and temporal convolution.

The reviewers' opinions diverge. R2 does not find any major flaws of the paper, while R1 expressed concerns in terms of experimental details, ablations, and missing comparison to the state-of-the-arts. R4 expressed a similar concern, while favoring the paper a bit more.

The AC agrees more with the senior reviewers (R1 and R4) that the paper misses its experimental comparison to the state-of-the-arts. In particular, the AC supports the statement from R4 that "Some SOTA performances are ignored selectively" to favor the proposed approach. The missing SOTA includes Slow-Fast as pointed out by R4, X3D as pointed out by R1, and more. For instance, X3D is able to obtain 80.4% top-1 accuracy on Kinetics-400, which is superior to 76.9% of this paper, but is being ignored in the paper. X3D that uses a different strategy to abstract temporal information than this paper is also superior in terms of the FLOPS: X3D gets 79.1% while using almost half of the computation the proposed approach is using. The authors responded by "X3D uses network architecture search strategy to discover the optimal setting for 3D CNNs", but this is a misleading statement as X3D does not use any neural architecture search method. The authors argue that their proposed approach might be able to also benefit X3D, but this has not been confirmed and we cannot judge since no quantitative results are provided. In addition, as mentioned, several other standard baselines such as Non-Local R101 (with 77.7% accuracy) and ip-CSN-152 (with 77.8% accuracy) performing better than the proposed approach are missing.

Overall, we find the experimental section of the submitted paper incomplete.